# The Association of Gender Role Attitudes and Risky Drinking: Changes in the Relationship between Masculinity and Drinking in Korean Young Men

**DOI:** 10.3390/ijerph192114271

**Published:** 2022-11-01

**Authors:** Joon-Yong Yang, Aeree Sohn

**Affiliations:** 1Institute of Health and Environment, Seoul National University, Seoul 08826, Korea; 2Department of Public Health, Sahmyook University, Seoul 01795, Korea

**Keywords:** problem drinking, gender role attitudes, Korean men, age group

## Abstract

Recently, younger men in Korean society tend not to view drinking as a proud feature of men who work. The relationship between gender role attitudes and high-risk drinking is expected to change accordingly. An online survey was conducted in January–February 2022, and the frequency of drinking, the amount of drinking, and traditional gender role attitudes, such as “men should be independent and women should take care of the children”, were measured. Participants were 786 men aged between 19 and 69 years. When comparing men in their 20s and 30s with those in their 40s to 60s, the younger men reported drinking relatively high amounts of alcohol in one place, and older men had a relatively high frequency of drinking. High-risk drinking was significantly higher in the older group. Gender role attitudes were more traditional in the older group and were a predictor of high-risk drinking in older men, but there was no significant relationship between the two variables in the younger age group. The results indicate that different generations have different perceptions of how men should drink. In addition to more detailed qualitative research on drinking motivation by generation, research on whether similar changes are occurring in other countries is needed.

## 1. Introduction

Problem drinking, drug abuse, smoking, overeating, and reckless driving are cited as representative health risk behaviors [1]. However, the status of drinking among these is slightly more special. The history of alcohol in human history is longer than that of any other addictive substance and is spread worldwide, regardless of the culture. It remains the only addictive substance that is not internationally regulated due to its unique sociocultural status [2]. According to the World Health Organization (WHO), drinking not only exacerbates most chronic and communicable diseases in individuals but also amplifies other health risk behaviors. Moreover, at a macrolevel, it has a profound impact on health inequality and economic inequality between underdeveloped and developed countries.

In Korea, drinking is a health risk behavior that needs to be managed more carefully. According to the National Health Statistics of the Ministry of Health and Welfare of Korea, as of 2020, the standardized high-risk drinking rate was 21.6% among Korean men (i.e., seven or more drinks more than twice a week) [3]. Between 2007 and 2019, the male smoking rate showed a marked decrease of 12.1% from 47.8% to 35.7%, while the monthly drinking rate decreased by only 4.4%. The mortality rate per 100,000 population due to alcohol-related diseases in Koreans remained unchanged at 9.6% between 2005 and 2018 [4]. As a result of estimating the socioeconomic cost of drinking, the social burden was greater than that of smoking and obesity, and an increasing trend was confirmed in the mid-2010s [5]. Few national comparative studies on drinking problems have included South Korea. However, a comparison of data from the National Epidemiologic Survey on Alcohol and Related Conditions with the Korean Epidemiologic Catchment Area showed that Americans had higher rates of alcohol abuse than Koreans (5.3% vs. 2.0%, respectively). Meanwhile, Koreans had a higher rate of alcohol dependence than Americans (5.1% vs. 4.4% respectively), and men had more drinking problems than women in both countries [6].

Regarding the higher drinking tendency in men compared with women, from a biological perspective, Walther et al. [7] noted gender differences in steroid secretion patterns, while Peters et al. [8] pointed out the association between testosterone and amygdala-OFC. However, the association between drinking behavior and masculinity has been found repeatedly in several studies [9,10,11,12,13]. Men participate in drinking for reasons such as coping with stress [9], showing off masculinity [10], and conforming to social norms [11]. However, drinking in East Asian cultures is understood to be closely related to men’s status and social activities at a more collective level [14,15]. According to Kim Min-hye et al. [5], Korean men’s perception is that it is only possible to reveal their feelings and form closer relationships through drinking parties and drinking alcohol. Therefore, drinking in Korea goes beyond the individual level, such as the acceptance of masculine norms, and is closely linked to masculinity at the collective level as a culture shared by male groups.

However, as masculinity is recognized differently due to generational changes in Korean society, the attitudes of men toward drinking are also changing [16,17]. For example, Min et al. [17] investigated the drinking motives and norms of 3000 Korean adults and stated that the drinking patterns of men in their 30s showed masculine characteristics. Unlike other groups, men in their 30s were more inclined to have good feelings toward drinking, one-shot experiences, bombshell experiences, and excessive drinking with co-workers. Super-masculine drinking behavior is relatively common in men in other countries. However, it is clearly distinct from the sociable and workplace-related drinking party-centered culture of men in their 50s, which is the main drinking pattern in Korean society. In contrast, Kang and Kim [16] found that middle-aged people still view drinking as a privilege and show off the amount of alcohol consumed, while the younger generation was relatively indifferent to the amount of alcohol consumed and they tended to relate their identity to drinking through alcohol brand selection. Additionally, young men who participated in the study clearly expressed their objection to the perception that drinking was masculine.

The change in the masculinity of young men in Korea, called “Lee Dae-Nam (20s Male)”, is the result of a decrease in the possibility of achieving social status due to the widening gap between the rich and the poor, a change in the workplace culture due to labor flexibility and frequent turnover, and the emergence of masculinity in consumption [18,19,20,21,22]. Young men who experience socioeconomic instability are no longer able to belong to the male patriarchal group based on their home or work, as in the past. Therefore, rather than learning traditional masculinity by following the older men, they resist the existing hegemonic masculinity to regain their lost position. Young men in Korea claim that they have superior masculinity compared with older men in two major ways. First, they identify themselves as fairer and more rational beings [18]. For them, displaying the amount of drinking or binge drinking need to be excluded as irrational acts that do not improve economic productivity. Second, they show off through consumption that their tastes are more sophisticated than those of the older generation [19]. Unlike in the past, the image of men became an object of consumption through cultural media such as movies; as a result, the younger generation envies the “aristocratic” drinking behavior of enjoying expensive alcohol rather than excessive drinking of cheap alcohol. A similar phenomenon was also observed in England’s young male population [23].

One way to confirm the assumption that the Korean younger generation no longer associates drinking with the traditional values of masculinity is to examine whether young men’s gender role attitudes are likely to lead to drinking. Gender role attitudes refer to an individual’s perception of whether a traditional gender role is positive or negative. The more traditional the gender role attitudes in men, the more they perceive the roles of men and women to be different, and more “masculine” (aggressive, proactive, and impulsive) behavior is affirmed [24]. If these assumptions are true, the relationship between traditional gender role attitudes and drinking in the younger generation would be weak or irrelevant, whereas the risk of drinking should increase in the older generation. This is because the younger generation no longer looks for the difference in drinking between men and women. Several previous studies have shown that men with traditional gender role attitudes drink more than men who do not follow traditional gender roles, and women with traditional gender role attitudes drink less than women who do not have these traditional gender attitudes [25,26,27,28]. However, few studies have examined whether the role of gender role attitudes can differ or change according to age in men.

Two major previous studies have examined the relationship between male gender role attitudes and drinking in Korea. Chung [29] examined the correlations among gender role attitudes, difficulties in work–family balance, and drinking behavior in 8692 adults using data from the Eighth Korean Welfare Panel. When marital status, employment status, and preschool children were controlled for, more positive attitudes towards men’s traditional gender role were associated with lower alcohol consumption and problematic drinking. However, because this study did not control for age, the tendency to reduce drinking due to a deterioration in health as age increases while having a positive attitude to traditional gender roles may have contributed to these results. Chang [30] conducted a path analysis using gender role attitudes as an independent variable while examining the factors related to problem drinking in 4891 older adults aged 65 and over, using data from the 13th Korean Welfare Panel. The results showed that traditional gender role attitudes increased problematic drinking through depression.

This study aimed to examine men’s drinking patterns in relation to age, gender role attitudes, and drinking behaviors. Based on previous empirical research, two hypotheses were examined. First, we hypothesized that drinking patterns would differ between the younger and older generations. Second, we hypothesized that traditional gender role attitudes would predict high-risk drinking; however, the size of the effect would differ by age.

## 2. Materials and Methods

### 2.1. Participants and Procedures

A separate survey was conducted for this study. The survey was named the “Gender Roles and Health Survey”. The main survey was conducted online from 24 to 26 January 2022. Additional investigations were conducted online for three days from 8 to 10 February 2022 by company Macromill Embrain (Seoul, Korea). Notably, during the survey period, the COVID-19 quarantine policy was eased in Korea. Specifically, except for wearing masks, patterns of public life reverted to those before the pandemic. However, it is likely that the psychological and social impact of the COVID-19 pandemic influenced participants’ drinking behavior. This may be reflected in their reports of drinking frequency and average amount of alcohol consumed in the past year. Therefore, caution is required when interpreting the results of this survey or comparing them with findings from before or after the height of the COVID-19 pandemic.

The study participants were directly recruited using an online panel owned by the research company, and the target population was adults between the ages of 19 and 69 across the country. To extract the sample, a random stratified sampling method was used considering the area of residence and age. Age was arranged to include a similar number by generation; in the case of gender, the number of participants was intentionally designated because it was a major variable in this study.

In total, 1331 participants (897 men and 434 women) were recruited, and the analysis was limited to annual drinkers who had had at least one drinking experience in 2021. Of these, 786 (87.6%) men were selected as participants. The average age of the participants was 43.98 years (SD = 13.50). The age group was largely divided into those in their 20s and 30s (*n* = 318, 40.5%) and 40s to 60s (*n* = 468, 59.5%), in consideration of the characteristics of masculinity in Korea. Of those in their 20s–30s, 24.5% were married or living with a partner, and 80.6% of the men in their 40s–60s were married or living with a partner. In terms of the monthly average household income, 10.4% of the participants were classified as low-income (around KRW2 million), 69.5% as middle-income (KRW201–207 million), and 20.1% as high-income (KRW701 million or more). The participants’ level of education was 19.2% high school graduates, 67.1% college and university graduates, and 13.7% with graduate school or higher. During the data cleaning process, insincere response patterns or excessive outliers were not found; therefore, the quality of the survey was judged to be good overall. Additional weights were not applied during the analysis.

### 2.2. Measures

#### 2.2.1. Outcome Variables: Drinking Behaviors

In this study, the frequency of drinking and the average amount of alcohol consumed were measured separately. Drinking frequency was assessed by asking “How often did you drink (in the past year)?” and was divided into five categories: less than once a month, about once a month, about 2–3 times a month, 2–3 times a week, and 4 or more times a week. The amount of drinking was arranged in units of standard glasses by asking, “How much alcohol did you usually drink in one place (in the past year)?”

High-risk drinking variables were constructed using the Korean Disease Control and Prevention Agency (KCDC) standards [3]. The KCDC classifies high-risk drinkers as those who consume more than seven standard drinks at a time, and who drink more than twice a week on average per month. We did not use indicators from the WHO or the National Institute on Alcohol Abuse and Alcoholism because the KCDC standards are more suitable for Koreans and have already been used in many Korean studies. Therefore, they are highly comparable.

#### 2.2.2. Independent Variables: Gender Role Attitudes

Gender role attitude (GRA) scales are used worldwide, especially in comparisons between countries, to measure an individual’s gender role values [31]. The GRA scale adopts a method of contrasting egalitarian gender role perceptions with traditional gender role perceptions, which assumes that the social perception of gender roles changes as women’s socioeconomic participation increases. Representative tools for measuring gender role attitudes include the Attitudes Toward Women Scale [32] and Sex Role Egalitarianism Scale. In few comparative studies of countries, such as surveys, the Eurobarometer, the European Social Survey, the International Social Survey Program, and the World Values Survey, related items have been used to confirm the level of gender equality in each country [33]. However, GRA-related measures have faced some criticism. First, due to social change, the perception of housework and women’s social participation has changed considerably, and the goal of gender equality has changed. Second, GRA-related scales have a strong gender-dichotomous nature and therefore do not capture attitudes toward multiple genders. Third, because of the inherent perception that men hold traditional gender roles, discussion about the changing gender roles of men has been limited [31]. For this reason, attempts to improve the GRA scales are ongoing; among them, the work of Vermeersch et al. [34] is representative.

In this study, the tool of Vermeersch et al. for measuring gender role attitudes [34] was used to translate the items that were subjected to exploratory factor analysis and additional tests for validity and reliability by Halimi et al. [31]. Since Vermeersch et al. [34] developed it as a tool for adolescents, its use in the general population may be questioned. However, compared with other gender role attitude scales for the general population, the format of the questions was not significantly different; therefore, this tool was selected with more emphasis on the merits of accepting criticism from the perspective of gender role change and undergoing statistical verification. The tool consists of 11 single-dimensional items that were translated into Korean and used in this study. The 11 items were responded to using a 5-point scale ranging from 1 (not very much) to 5 (very much). Higher scores indicate more support for traditional gender roles.

An exploratory factor analysis was performed on the measured data to determine whether the use of the gender role attitudes scale was appropriate. During orthogonal rotation, the varimax method was adopted, and factor analysis was performed using the principal axis decomposition method. The sample adequacy measurements and Bartlett’s sphericity verification results before the exploratory factor analysis were all good, and the MSA value was 0.860 or higher, showing no abnormalities. Two factors could be distinguished according to the Kaiser criterion, which determines whether a factor with an eigenvalue of 1 or more is a meaningful separate factor. As a result of verifying the internal fit of all 11 items, good values were also obtained (Cronbach’s α = 0.876). Accordingly, the 11 items were composed of a single dimension, in line with the results of the original study that developed the tool.

#### 2.2.3. Control Variables: Norms

In many studies on drinking norms, injunctive and descriptive norms are used interchangeably [35]. When injunctive and descriptive norms are clearly distinguished, the influence of descriptive norms on drinking behavior is greater than that of injunctive norms [36,37]. Many studies have shown that drinking norms play an important role in classifying masculinity as part of masculinity norms [11]. According to Chung et al. (2012) and Allen Rose et al. (2020), a close relationship exists between drinking norms and traditional gender role attitudes in Korea, leading to more drinking in men [38,39]. In particular, it is known that young men are greatly affected by drinking norms because young men, such as college students, have relatively more organized and explicit drinking norms than other population groups [40]. Therefore, in this study, we included young men as a control variable, fearing that drinking norms would function as a confounding variable that could affect both gender role attitudes and high-risk drinking.

To confirm the injunctive and descriptive norms of drinking, one of the normative items that showed the greatest influence on drinking behavior was selected in a study related to drinking norms in Koreans [37,41].

Injunctive drinking norms were measured based on tolerance of drinking. Participants were asked “What do you think about drinking and getting drunk?” Reponses were rated on a 5-point scale ranging from “not okay at all” to “very good.” Descriptive drinking norms gauged friends’ drinking behavior, asking “How often do you think your friends drink alcohol?” Six categories were selected: never drank, less than once a month, once a month, 2–4 times a month, 2–3 times a week, and 4 or more times a week.

#### 2.2.4. Sociodemographic Variables

Age, education level, marital status, household income, and smoking status were selected as sociodemographic variables. Age was classified as international age, and education level was classified as lower than high school graduate, junior college graduate, college graduate, or higher than graduate school. For the classification of marital status, married and cohabiting were combined, and the rest were separated. Household income was divided into eight categories: none, ~KRW2 million, KRW2.01–3 million, KRW3.01–4 million, KRW4.01–5 million, KRW5.01–6 million, KRW6.01–7 million, and more than KRW7.01 million. Smoking was classified into three categories: no smoking at all, quit smoking in the past, and smoking.

### 2.3. Data Analysis

In this study, a binary logistic regression analysis was performed because the dependent variables had values of 0 and 1. Before the logistic regression, relationships within gender role attitudes variables were identified using correlation analysis. All analyses were performed using IBM SPSS 20.

## 3. Results

### 3.1. Drinking Behaviors

Table 1 shows the results for the drinking behaviors of the two age groups. For participants in their 20s and 30s, the frequency of drinking was highest in the order of 2–3 times a month (28.3%), less than once a month (23.9%), and 2–3 times a week (23.3%). In contrast, for participants in their 40s to 60s, the sum of 2–3 times a week (31.2%) and 2–3 times a month (26.5%) exceeded the majority (57.7%). The average amount drunk at one place based on standard drinks was higher in the younger age group: 4.34 drinks (SD = 2.44) for participants in their 20s and 30s and 3.97 drinks (SD = 2.10) for those in their 40s to 60s. As a result of the chi-square independence test and independent sample *t*-test, the frequency of drinking was significantly higher in the older age group, while the average amount drunk was higher in the lower age group at the 95% confidence level.

The high-risk drinking rate was 24.5% in participants in their 20s and 30s, and 38.0% in those in their 40s to 60s, a difference of 13.5%. The chi-square test indicated the difference was significant at a confidence level of 99.9%.

### 3.2. Gender Role Attitudes

Table 2 shows the differences in gender role attitudes by age group, item score, and total scale score. In particular, for Items 3 (M_younger_ = 2.33, M_older_ = 3.11) and 5 (M_younger_ = 2.72, M_older_ = 3.51), the differences between the generations are remarkable. In both age groups, the strongest endorsement was for Item 1 (M_younger_ = 3.06, M_older_ = 3.60), referring to “A boy behaving like a girl is disturbing” and the lowest was for Item 8 (M_younger_ = 1.87, M_older_ = 2.13), referring to whether a woman performing a typically masculine job was unfeminine. When the average of all questions was calculated to create a total score, the gender role attitudes score for those in their 20s and 30s was 2.47 (SD = 0.77), and that for those in their 40s to 60s was 2.90 (SD = 0.61). The difference was 0.43 points higher for the older generation, and this difference was significant at a confidence level of 99.9%.

Table 3 shows the results of calculating the Pearson’s correlation coefficients for the relationships among the items assessing gender role attitudes. The correlation coefficients ranged between 0.210 and 0.632.

Close relationships were found among the items. Considering this result, the results of the previous exploratory factor analysis and the evaluation of internal consistency reliability, the gender role attitudes scale was considered to be a unidimensional scale that assesses traditional attitudes about gender roles. Higher scores indicate more traditional attitudes.

### 3.3. Binary Logistic Regression Analyses Predicting High-Risk Drinking across Age Groups

Logistic regression analysis of the factors influencing high-risk drinking according to age group revealed significant differences in age, education level, and gender role attitudes (Table 4). For those in their 20s and 30s, the older the age, the more the smoking, and the higher the compliance with the drinking norms, the higher the probability of being a risky drinker. However, among those in their 40s to 60s, the likelihood of high-risk drinking was high in those who were younger, educated, smoked, conformed to drinking norms, and had traditional gender role attitudes. In the case of age, it can be inferred that the highest rate of risky drinking tends to occur around 40 years of age. Smoking and drinking norms showed relatively consistent positive correlations with risky drinking regardless of age. However, education level and gender role attitudes were correlated with risky drinking only in the relatively older age group.

## 4. Discussion

### 4.1. Drinking Behaviors

This study examined whether gender role attitudes could have a significant effect on high-risk drinking among Korean men when various demographic variables and drinking norms were controlled for. In the process, the generations were divided into two age groups: those in their 20s and 30s, and those in their 40s to 60s. More attention was paid to the difference in the influence of gender role attitudes. While the results indicated that Korean men in the older age group drank more frequently, the average amount of alcohol consumed in one place was higher in the younger age group. Among annual drinkers, high-risk drinkers accounted for 38% of men in their 40s to 60s, 13.5% more than 24.5% of men in their 20s and 30s. Therefore, high-risk drinking in the older age group was more closely related to the frequency of drinking than to the amount of alcohol consumed on each occasion of drinking.

In this study, consistent with Chung et al. (2012) and Windle et al. (2008), an inverted U-shaped curve trend of high-risk drinking according to age was found [38,42]. In those in their 20s and 30s, the probability of belonging to the high-risk drinking group increased with age (95% CI = 1.043–1.206), whereas the probability of belonging to the high-risk drinking group decreased in those in their 40s to 60s (95% CI = 0.938–0.990). In older age groups, a decrease in high-risk drinking with increasing age has also been observed in Russia and Japan [43,44].

Education level was found to be a predictor of high-risk drinking in the older group (95% OR = 0.522–0.857) but not in the younger generation. As summarized by Huerta and Borgonovi [45], contradictory claims have been drawn regarding the effect of the level of education on drinking behaviors. However, in Korea, many studies have shown that the higher the education level, the lower the rates of alcohol consumption and problem drinking [5,46,47]. This may be because the higher the education level, the more the adverse health effects of drinking are recognized, and the higher the socioeconomic status, the more people have to lose from problem drinking [45]. However, the higher the education level, the more likely individuals are to belong to groups that engage in drinking, such as college and work groups [9,40,45]. The buffering effect of marital status and household income did not appear to be significant predictors of high-risk alcohol consumption. In contrast, smoking and drinking norms were positively correlated with high-risk drinking in both age groups. These results confirm that smoking [44,48] and drinking norms are strong predictors of high-risk drinking [5,41].

### 4.2. Traditional Gender Role Attitudes

When examining gender role attitudes, the older group had higher mean scores than the younger group for all 11 measurement items and the total score at a 95% confidence level. Item 1 (“I find it disturbing if a boy behaves like a girl”) had the highest score for both those in their 20s–30s and 40s–60s, and Item 8 (“If I heard a woman was a mason or a roofer, I would doubt whether she was ‘feminine’”) had the lowest score. When ranking the items from the highest mean score to the lowest by age group, the younger age group’s rankings were Items 1, 5, 11, 4, 7, 9, 10, 2, 3, 6, and 8, and the older group’s rankings were Items 1, 5, 4, 3, 11, 2, 10, 6, 7, 9, and 8. Items 7 (“Only slim girls are attractive to boys”) and 9 (“A man should avoid being dependent on others”) were ranked more highly by the younger group than by the older group. The item with the highest ranking was Item 3 (“There is definitely something wrong with a boy practicing ballet as a hobby”). Considering the items’ rankings, the relatively younger generation scored more traditionally regarding what the position of a woman should be, whereas the older generation showed a more traditional view of what a man position should be.

Men’s traditional gender role attitudes were a predictor of high-risk drinking in the older age group, which is consistent with previous studies [25,26,27,28,30]. This may be due to the traditional tendency of older generations to choose drinking as one of the ways in which men and women differ. At a confidence level of 95%, the odds ratio of traditional gender role attitudes was between 1.054 and 2.199. However, in the younger age group in their 20s and 30s, the 95% odds ratio of traditional gender role attitudes was between 0.695 and 1.622, which did not significantly predict high-risk drinking.

Piotrowski et al. [49] showed a change in gender role attitudes by cohort in Japan. On the one hand, Koreans in their 20s and 30s born between 1984 and 2003 have benefited directly from globalization. On the other hand, they have different socioeconomic characteristics from previous generations who started their social life after the International Monetary Fund’s economic aid and were accustomed to chronic economic crises and income polarization, leading to a social life during a period of economic growth [18,19]. Using economic growth as a justification mechanism, this generation could not fully fulfill the East Asian “salary-man masculinity” concept of owning women through money [50]. Therefore, there is relatively little sense of utility in strengthening male solidarity by expanding “masculine capital” as a gendered social capital through drinking at work, such as “hoe-sik (dining parties)” [51]. Thus, as Min et al. [17] pointed out, East Asian drinking, which confirms the hierarchy and order of men, was confirmed only in the older Korean male group, and younger men drink for the purposes of promoting friendship in an intimate in-group and escaping from daily life. However, identity consumption trends are added to the drinking of young men, and the selection of alcohol brands has become more important than the amount of alcohol consumed. This means that they no longer drink as “men”, but to express the “me” [16,23]. It is possible that the reason why the relationship between traditional gender role attitudes and risky drinking in the younger generation was not significant was because younger men do not perceive drinking as a behavior that differentiates young men and women in Korea. The relationship between men and drinking is being reconstructed.

### 4.3. Limitations

This study has several limitations that deserve mentioning. First, although the reliability and validity of the developed gender role attitudes scale was verified, it has not been confirmed whether the reliability and validity were maintained even after being translated into Korean and used for all Korean males. Second, a standardized alcohol consumption questionnaire, such as the AUDIT-10, was not used. As the amount and frequency of drinking were self-reported, reports of drinking behavior can be arbitrary and recall bias may have affected responses to the items. Third, drinking norms were controlled as variables that could be correlated with both high-risk drinking and gender role attitudes, but other key drinking-related variables such as drinking motives were not sufficiently controlled for. Fourth, these data are cross-sectional, so it is not possible to determine causality. Fifth, men’s unique psychological problems, such as male-specific depression symptoms that can affect men’s drinking, were not controlled for.

Another consideration is the close relationship between masculinity and drinking behavior, and that high-risk drinking itself may have the effect of reinforcing gender role attitudes; thus, attention should be paid to the interpretation of the relationships between the variables. Furthermore, the gender-dichotomous characteristic of the gender role attitudes scale itself remains, and it is regrettable that it did not explain the differences within men according to sexual and gender identity, such as gay, bisexual, or transgender men. Finally, although gender role attitudes were the focus of our study, it is important to also assess the extent to which gender role norms have changed over the past 70 years, and to assess the participants’ perceptions of such changes.

## 5. Conclusions

This study confirmed that young Korean men in their 20s and 30s consumed more drinks than older men on each occasion of drinking, while older men in their 40s to 60s drank more frequently. In addition, in younger men, gender role attitudes were not a predictor of high-risk drinking, whereas in older men, the more traditional the gender role attitudes, the higher the probability of belonging to the high-risk drinking group. Of course, a change in gender role attitudes toward egalitarianism is welcome from both a gender equality perspective and a drinking risk perspective, but the expectation that these changes will reduce male drinking behavior may no longer be meaningful. Young men in Korea tend to drink in a way that is unrelated to their gender role attitudes. According to previous studies [16,17] and the overall results of this study, the change from normative, sociable, and male solidarity drinking to non-normative, impulsive, and personal identity-centered drinking is likely to occur. In the future, by studying how variables such as gender role attitudes and gender role norms relate to drinking norms and motives, this assumption suggested in this study can be confirmed.

## Figures and Tables

**Table 1 ijerph-19-14271-t001:** Drinking behaviors across age groups.

Age Group				
Drinking Frequency and Quantity	Total (*n* = 786) %	20s–30s (*n* = 318) %	40s–60s (*n* = 468) %	$x^{2}$ or t (p)
Drinking Frequency				$x^{2}=30.832$ (<0.001)
Less than once per month	19.3	23.9	16.2	
Once per month	13.5	17.9	10.5	
Two or three times per month	27.2	28.3	26.5	
Two or three times per week	28.0	23.3	31.2	
Four or more times per week	12.0	6.6	15.6	
Drinking quantity on each occasion (# of standard drinks)				
M (SD)	4.12 (2.25)	4.34 (2.44)	3.97 (2.10)	t = 2.223 (0.027)
High-risk drinker	32.6	24.5	38.0	$x^{2}=15.726$ (<0.001)

**Table 2 ijerph-19-14271-t002:** Items and total scores across age groups for gender role attitudes.

Age Group				
Gender Role Attitudes: Traditional	Total (*n* = 786) M (SD)	20s–30s (*n* = 318) M (SD)	40s–60s (*n* = 468) M (SD)	t (p)
(1)	3.38 (1.13)	3.06 (1.26)	3.60 (0.97)	t = 6.471 (<0.001)
(2)	2.58 (0.94)	2.35 (1.00)	2.74 (0.87)	t = 5.636 (<0.001)
(3)	2.79 (1.15)	2.33 (1.13)	3.11 (1.05)	t = 9.705 (<0.001)
(4)	2.98 (1.12)	2.59 (1.16)	3.25 (1.01)	t = 8.267 (<0.001)
(5)	3.19 (1.14)	2.72 (1.24)	3.51 (0.94)	t = 9.704 (<0.001)
(6)	2.50 (0.99)	2.25 (1.02)	2.67 (0.94)	t = 5.838 (<0.001)
(7)	2.58 (1.10)	2.47 (1.19)	2.66 (1.03)	t = 2.272 (0.023)
(8)	2.03 (0.98)	1.87 (1.01)	2.13 (0.95)	t = 3.580 (<0.001)
(9)	2.56 (1.00)	2.46 (1.09)	2.63 (0.93)	t = 2.294 (0.022)
(10)	2.59 (1.09)	2.42 (1.13)	2.71 (1.04)	t = 3.574 (<0.001)
(11)	2.79 (1.06)	2.61 (1.12)	2.92 (1.00)	t = 3.992 (<0.001)
Total	2.73 (0.71)	2.47 (0.77)	2.90 (0.61)	t = 8.427 (<0.001)

(1) I find it disturbing if a boy behaves like a girl. (2) It is in everyone’s best interest if the man makes the decisions within the family. (3) There is definitely something wrong with a boy practicing ballet as a hobby. (4) I find it disturbing if a girl behaves like a boy; (5) There is definitely something wrong with girls who use foul language. (6) It is in everyone’s best interest if a woman stays at home and does not go to work once there are children. (7) Only slim girls are attractive to boys; (8) If I heard a woman was a mason or a roofer, I would doubt whether she was “feminine”. (9) A man should avoid being dependent on others. (10) A man without self-confidence is an idiot, (11) A real man doesn’t give in; he fights back.

**Table 3 ijerph-19-14271-t003:** Pearson’s correlations among the 11 gender role attitudes.

Item	(1)	(2)	(3)	(4)	(5)	(6)	(7)	(8)	(9)	(10)	(11)
(1)	1										
(2)	0.353 **	1									
(3)	0.596 **	0.502 **	1								
(4)	0.632 **	0.414 **	0.604 **	1							
(5)	0.451 **	0.400 **	0.521 **	0.541 **	1						
(6)	0.274 **	0.472 **	0.444 **	0.364 **	0.395 **	1					
(7)	0.217 **	0.345 **	0.327 **	0.292 **	0.270 **	0.314 **	1				
(8)	0.210 **	0.395 **	0.394 **	0.319 **	0.264 **	0.422 **	0.432 **	1			
(9)	0.259 **	0.463 **	0.397 **	0.296 **	0.289 **	0.385 **	0.373 **	0.480 **	1		
(10)	0.330 **	0.391 **	0.430 **	0.337 **	0.374 **	0.359 **	0.360 **	0.361 **	0.532 **	1	
(11)	0.324 **	0.426 **	0.409 **	0.352 **	0.350 **	0.367 **	0.346 **	0.315 **	0.469 **	0.584 **	1

** *p* < 0.01.

**Table 4 ijerph-19-14271-t004:** Logistic regression analyses predicting high-risk drinking across age groups.

Demographic Variable	20s–30s (*n* = 318)			40s–60s (*n* = 468)		
	OR	Lower 95% CI	Higher 95% CI	OR	Lower 95% CI	Higher 95% CI
Age	1.122 **	1.043	1.206	0.964 **	0.938	0.990
Education	1.130	0.772	1.654	0.669 **	0.522	0.857
Marital status	1.083	0.532	2.203	1.696	0.939	3.064
Household income	0.841	0.617	1.148	1.050	0.832	1.326
Smoking	1.798 ***	1.285	2.517	1.464 **	1.115	1.924
Descriptive norms	1.527 **	1.119	2.082	1.400 **	1.117	1.755
Injunctive norms	2.350 ***	1.649	3.349	2.365 ***	1.852	3.019
Gender role attitudes: Traditional	1.062	0.695	1.622	1.522 *	1.054	2.199
$\mathrm{Nagelkerke}’s\;R^{2}$	0.338			0.288		
−2 log-likelihood	272.295			510.355		

OR, odds ratio; CI, confidence interval. * *p* < 0.05, ** *p* < 0.01, *** *p* < 0.001.

## Data Availability

Any queries regarding the data used in this study may be directed to the corresponding author. The dataset used in the present study is available on reasonable request.

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
