# Peer review of "The Association of Gender Role Attitudes and Risky Drinking: Changes in the Relationship between Masculinity and Drinking in Korean Young Men"

_ijerph, 2022, doi:10.3390/ijerph192114271_

Round 1
Reviewer 1 Report (Previous Reviewer 1)
The authors have revised the manuscript according to the suggestions. Bi further edits are Neides.
This manuscript is a resubmission of an earlier submission. The following is a list of the peer review reports and author responses from that submission.
Round 1
Reviewer 1 Report
The authors have written an exciting paper on drinking behavior in Korean men, looking more closely at age differences and the relationship to gender role norms.
Title:
The title has to be changed, because it is not about an effect of an intervention, but about an association. Therefore, please change "Effect" to "Association".
Abstract:
- Please state the sample size.
- Please report, if the data is available, whether it is cismale or also other gender identities (e.g., trans men).
- Try to explain briefly what was meant and measured by gender role attitudes. There are many instruments that go in this direction and therefore it would be helpful if the reader could understand directly in the abstract what is meant by it.
Introduction:
- As the authors try to make the point, that alcohol use in Korea needs to be addressed more carefully, it seems odd that the authors highlight that 73% of Korean men drank alcohol in the last month. This is absolutely uninformative, since drinking alcohol once a month is not at all hazardous to health. Therefore, it would be much more necessary to use the prevalences of alcohol use disorders for Korea as well as other countries in comparison.
Grant, B. F., Goldstein, R. B., Saha, T. D., Chou, S. P., Jung, J., Zhang, H., ... & Hasin, D. S. (2015). Epidemiology of DSM-5 alcohol use disorder: results from the National Epidemiologic Survey on Alcohol and Related Conditions III. JAMA psychiatry, 72(8), 757-766.
Kendler, K. S., Ohlsson, H., Sundquist, J., & Sundquist, K. (2016). Alcohol use disorder and mortality across the lifespan: a longitudinal cohort and co-relational analysis. JAMA psychiatry, 73(6), 575-581.
Grant, B. F., Chou, S. P., Saha, T. D., Pickering, R. P., Kerridge, B. T., Ruan, W. J., ... & Hasin, D. S. (2017). Prevalence of 12-month alcohol use, high-risk drinking, and DSM-IV alcohol use disorder in the United States, 2001-2002 to 2012-2013: results from the National Epidemiologic Survey on Alcohol and Related Conditions. JAMA psychiatry, 74(9), 911-923.
- Although the focus of the paper is on sociocultural reasons for problem drinking behavior, it should at least be added that biologically-based theories also exist.
Walther, A., Rice, T., Kufert, Y., & Ehlert, U. (2017). Neuroendocrinology of a male-specific pattern for depression linked to alcohol use disorder and suicidal behavior. Frontiers in psychiatry, 7, 206.
Peters, S., Jolles, D. J., Van Duijvenvoorde, A. C., Crone, E. A., & Peper, J. S. (2015). The link between testosterone and amygdala–orbitofrontal cortex connectivity in adolescent alcohol use. Psychoneuroendocrinology, 53, 117-126.
- I want to make the authors a big compliment for the exciting delineation of drinking motivation in Korean men. Very exciting.
- It is not entirely clear, why the authors chose the concept of endorsement of traditional the gender role attitudes. For example the BSRI is based on the rather outdated Gender Role Identity Paradigm instead newer concepts such as the endorsement of traditional masculinity ideologies (MRNI or CMNI) are based on the Gender Role Strain Paradigm. Since only men were studied, it would have been fitting to use such male-specific instruments. Please elaborate on that.
- The introduction ends abruptly and the method section is presented without a transition. Please create a better transition and at best formulate clear study hypotheses at the end of the introduction.
- Although it is clear from the Introduction that the authors followed the example of previous large studies in Korea, it is regrettable that more male-specific instruments, such as the assessment of male-specific depression symptoms, were not included. This should at least be clearly emphasized in the limitations. Because it could be clearly shown that for different health behaviors in men, male-specific depression symptoms can clarify much of the total variance.
For example:
Eggenberger, L., Komlenac, N., Ehlert, U., Grub, J., & Walther, A. (2022, June 6). Association Between Psychotherapy Use, Sexual Orientation, and Traditional Masculinity Among Psychologically Distressed Men. Psychology of Men & Masculinities. Advance online publication. http://dx.doi.org/10.1037/men0000402
Walther, A., Eggenberger, L., Grub, J., Ogrodniczuk, J. S., Seidler, Z. E., Rice, S. M., ... & Ehlert, U. (2022). Examining the role of traditional masculinity and depression in men's risk for contracting COVID-19. Behavioral Sciences, 12(3), 80.
Methods:
- Please explain possible influences of the COVID-19 pandemic on drinking behavior, as this global crisis exerts influences on many psychopathological behaviors and we should be aware of this, so this should be briefly described at least in the methods section that this data was conducted during a specific time of the pandemic. As an example, a recent article can be used, which also does this:
Status Loss Due to COVID-19, Traditional Masculinity, and Their Association With Recent Suicide Attempts and Suicidal Ideation (Walther et al., 2022; Psychology of Men and Masculinities):
The survey period took place in the midst of the third wave of COVID-19 infections in German-speaking Europe. While in Germany between 6,000 and 29,000 daily new COVID-19 infections were reported during this period, in Austria and Switzerland, the 7- day averages were around 2,000–3,000 new infections per day. By March 15, 2021, all countries were in the midst of the third wave of infection, and uncertainty was again high as to how severe this wave would become. Toward the end of April, it became apparent that the peak of the infection wave had been reached in all three countries. Thus, a relatively uniform picture of the current threat posed by the COVID-19 pandemic emerges for all participants, as it was a 1.5- month survey period from March 15, 2021 to April 28, 2021, representing a relatively short survey period, during which the pandemic situation remained steady.
- Since retrospective data over a whole year tend to be biased, it should be mentioned as a limitation that no standardized alcohol consumption questionnaire such as the AUDIT-10 four was used to record alcohol consumption, but that self-generated items were used.
- Furthermore, no indicators of reliability or validity are provided leaving the question, whether the alcohol-items really measure, what the authors are suggesting.
- The statistical analysis should consider control for multiple testing as many comparisons and correlations are examined. This might clearly change the results of the present study. Therefore, I would review a revised version of the manuscripts results and discussion section at a later stage in the review process.
Thank you for giving me the opportunity to review this very interesting article and I wish the authors a lot of success in their research endeavors.
Reviewer 2 Report
It is Interest subject and important findings on generation difference. However few things need to be added.
1. There is any difference between men and female role through the whole manuscript. Since the title shows gender role attitudes, it should be mentioned at least in somewhere, such as introduction, conclusion, or limitation of the study. Otherwise title should be changed as generation difference instead of gender role.
2. Abstract should be ended with future researches.
3. It should be more focused on Korean young men.